# Microglial Cytokines Induce Invasiveness and Proliferation of Human Glioblastoma through Pyk2 and FAK Activation

**DOI:** 10.3390/cancers13246160

**Published:** 2021-12-07

**Authors:** Rebeca E. Nuñez, Miguel Mayol del Valle, Kyle Ortiz, Luis Almodovar, Lilia Kucheryavykh

**Affiliations:** 1Department of Biochemistry, Universidad Central del Caribe, Bayamón, PR 00956, USA; rebeca.nunez@uccaribe.edu; 2Department of Surgery, Neurosurgery Section, Medical Sciences Campus, University of Puerto Rico, San Juan, PR 00936, USA; miguel.mayol@upr.edu (M.M.d.V.); kyle.ortiz1@upr.edu (K.O.); 3HIMA San Pablo Hospital, Caguas, PR 00725, USA; ljalmodovar@himapr.com

**Keywords:** glioblastoma, microglia, tumor microenvironment, Pyk2, FAK, invasion, proliferation, cell signaling, primary human cell lines, cytokines

## Abstract

**Simple Summary:**

Microglia infiltrate most gliomas and have been demonstrated to promote tumor growth, invasion, and treatment resistance. To develop improved treatment methods, that take into consideration the supporting role of microglia in tumor progression, the functional and mechanistic pathways of glioma–microglia interactions need to be identified and experimentally dissected. Our recent studies and literature reports revealed the overexpression of Pyk2 and FAK in glioblastomas. Pyk2 and FAK signaling pathways have been shown to regulate migration and proliferation in glioma cells, including microglia-promoted glioma cell migration. However, the specific factors released by microglia that modulate Pyk2 and FAK to promote glioma invasiveness and proliferation are poorly understood. The aim of this study was to identify key microglia-derived signaling molecules that induce the activation of Pyk2- and FAK-dependent glioma cell proliferation and invasiveness.

**Abstract:**

Glioblastoma is the most aggressive brain tumor in adults. Multiple lines of evidence suggest that microglia create a microenvironment favoring glioma invasion and proliferation. Our previous studies and literature reports indicated the involvement of focal adhesion kinase (FAK) and proline-rich tyrosine kinase 2 (Pyk2) in glioma cell proliferation and invasion, stimulated by tumor-infiltrating microglia. However, the specific microglia-released factors that modulate Pyk2 and FAK signaling in glioma cells are unknown. In this study, 20 human glioblastoma specimens were evaluated with the use of RT-PCR and western blotting. A Pierson correlation test demonstrated a correlation (0.6–1.0) between the gene expression levels for platelet-derived growth factor β(PDGFβ), stromal-derived factor 1α (SDF-1α), IL-6, IL-8, and epidermal growth factor (EGF) in tumor-purified microglia and levels of p-Pyk2 (Y579/Y580) and p-FAK(Y925) in glioma cells. siRNA knockdown against Pyk2 or FAK in three primary glioblastoma cell lines, developed from the investigated specimens, in combination with the cytokine receptor inhibitors gefitinib (1 μM), DMPQ (200 nM), and burixafor (1 μM) identified EGF, PDGFβ, and SDF-1α as key extracellular factors in the Pyk2- and FAK-dependent activation of invadopodia formation and the migration of glioma cells. EGF and IL-6 were identified as regulators of the Pyk2- and FAK-dependent activation of cell viability and mitosis.

## 1. Introduction

Glioblastoma (GBM) is the most common and malignant adult brain tumor, with an incidence close to 10 per 100,000 people per year worldwide [1,2]. Despite recent advances in health care standards, it has a poor prognosis, with a median survival of approximately 15 months [1]. The ability of glioma cells to restrict absorption of therapeutic compounds [3], uncontrolled proliferation, rapid invasion into the surrounding brain tissue [4], and a supportive tumor microenvironment [5,6] lays the basis for glioma resistance to treatment and fast relapse after surgical resection.

It has been demonstrated that the tumor microenvironment plays a vital role in glioma progression and treatment resistance [5,7,8,9,10,11]. Microglia and infiltrating macrophages account for up to one third of the glioma mass and represent important components of the immunosuppressive tumor microenvironment [12,13]. Indeed, tumor-infiltrating microglia have been associated with a higher tumor grade and a worse patient outcome and survival [14,15]. Under the influence of glioma, microglia are recruited to the tumor site and secrete cytokines, promoting glioma cell proliferation, invasion, and migration [14,16,17,18].

Tumor-activated microglia are directly involved in the degradation of the extracellular matrix (ECM). This process is mediated by the secretion of metalloproteases (MPPs) and is a critical mechanism for the expansion of tumors into the surrounding brain parenchyma [5,7,18]. Other microglial-secreted factors, such as transforming growth factor β1 (TGFβ1) [19], epidermal growth factor (EGF) [14], IL-10 [20], or stress-inducible protein 1 (STI1) [21], contribute to increased GBM invasiveness. Such factors appear to activate intracellular signaling pathways in glioma cells to enhance their invasiveness and proliferation. Recent studies from our lab [10] point to microglia-driven glioma cell invasiveness through the proline-rich tyrosine kinase 2 (Pyk2) mechanism.

Pyk2 and its relative, focal adhesion kinase (FAK), are non-receptor protein tyrosine kinases from the FAK protein family. These kinases function as signaling effectors through the stimulation of cell proliferation, migration, and survival pathways in astrocytes, fibroblasts, epithelial cells, and glioma cells [22,23,24]. Specifically, FAK was shown to be involved in glioma cell proliferation [24,25,26], while increased activity of Pyk2 correlates with invasion by glioma cells [10,27,28]. Studies by Lipinski et al. [24] showed that inhibition of both Pyk2 and FAK increases survival in mice with glioma cell xenografts. These kinases are activated by a range of transmembrane receptors (i.e., integrins and growth factor-, G protein-coupled-, and cytokine receptors) in various cell types, including neurons and glioma cells [24,29]. Altogether, these studies suggest that Pyk2 and FAK are key mediators of microglia-stimulated glioma cell migration and invasion. However, the specific cytokines or factors released by microglia that modulate Pyk2 and FAK to promote the invasiveness and proliferation of glioma are poorly understood.

The purpose of this study was to identify key cytokines and chemokines released by tumor-infiltrating microglia that induce the activation of Pyk2- and FAK-dependent glioma cell proliferation and invasiveness. Through an analysis of 20 human GBM specimens and three primary human glioblastoma cultures generated from three of these specimens, we determined that both Pyk2 and FAK signaling contribute significantly to glioma cell proliferation and dispersal. Our studies suggest that stromal-derived factor 1α (SDF-1α), EGF, platelet-derived growth factor β (PDGFβ), and IL-6 are key microglia-derived mediators of Pyk2 and FAK activation in primary human glioma cell lines, leading to increased invasion and proliferation. With these findings, we were able to dissect the functional role of microglia in glioma tumor progression and identify target candidates for the therapeutic disruption of glioma–microglia interactions.

## 2. Materials and Methods

### 2.1. Clinical Samples

Human resected glioblastoma tumor (grade IV GBM, WHO) specimens were obtained at the University District Hospital Rio Piedras Medical Center and the HIMA San Pablo Hospital, Puerto Rico, in accordance and with use of informed consent approved by the Institutional Review Board (IRB) Human Research Subject Protection Office (protocol #2012-12B). Inclusion criteria: ≥21 years of age and CNS neoplasia diagnosed by neuroimaging techniques. Grade IV GBM was confirmed by pathology analysis. Tissue samples, ~1.5–2.0 cm^3^ in volume, were separated from the resected tumor mass immediately after surgical dissection. Within 1 h of removal, the resected tissues were used for western blot and RT-PCR analyses, as well as for the preparation of primary glioblastoma cultures.

### 2.2. Glioblastoma and Microglia Purification from Tumor Specimens

Glioblastoma cells were purified from homogenized tissue using Percoll (Sigma-Aldrich, St. Louis, MO, USA, cat. #E0414) in concentration layers of 30%, 37%, and 70%. Following this procedure, the glioma fraction was collected from the top, and the distinct white ring of microglial cells was collected at the interface between the 37% and 70% Percoll layers. Both glioma and microglial fractions were used for further analysis.

### 2.3. Real-Time RT-PCR

Gene expression levels in glioma and microglial cells purified from tumor specimens were determined by RT-PCR analysis. Expression of the PTK2 and PTK2B genes encoding FAK and Pyk2, as well as the genes encoding EGFR, PDGFRα6, PDGFRβ, NGFR, IL-6R, CXCR1 (also known as IL-8R), CXCR4 (also known as SDF-αR), and CCR5 were analyzed in glioma cells. Expression of NGF, EGF, PDGFα, PDGFβ, IL-6, IL-8, CCL5, and CXCL12 (also known as SDF-1α) were analyzed in microglia. RNA was extracted from cell pellets using the RNeasy Plus Mini Kit (Qiagen GmbH, Hilden, Germany, cat. #74134), following the manufacturer’s protocol. RNA quality and concentration were quantified spectrophotometrically with the NanoDrop 1000 spectrophotometer (Thermo Scientific, Waltham, MA, USA). Complementary DNA (cDNA) was reverse-transcribed from 1 µg of total RNA using the iScript cDNA synthesis kit (Bio-Rad, Hercules, CA, USA, cat. #1708890). Asymmetrical cyanine dye SYBR green qRT-PCR gene expression assays were performed using SsoAdvanced Universal SYBR green Supermix (Bio-Rad, cat. #725271) with 50 nM Hs_RTK2 and Hs_RTK2B primers (Qiagen GmbH, cat. #QT00057687 and #QT00073402, respectively), as well as primers for the genes encoding NGFR, EGFR, PDGFRβ, CXCR1, CCR5 (Qiagen GmbH, cat. #QT 00056756, QT00085701, QT00082327, QY00212919, and QT01336601, respectively), PDGFRα, IL-6R, and CXCR4 (also known as SDF-1αR) in the glioma fraction, and the genes encoding NGF, EGF, PDGFα, PDGFβ, IL-6, IL-8, CCL5 (Bio-Rad, cat. #10025636), and CXCL12 (also known as SDF-1α, Bio-Rad, cat. #10041595) in the microglial fraction. A list of primers is provided in Appendix A. Amplification was carried out in a CFX96 Touch real-time PCR detection system (Bio-Rad). Mean fold changes in expression of the target genes were calculated using the comparative CT method (relative expression units, 2–ΔΔCt). All data were controlled for the quantity of input RNA by the GAPDH forward (5′-CTGGGCTACACTGAGCACC-3′) and reverse (5′-AAGTGGTCGTTGAGGGCAATG-3′) primers (Integrated DNA Technologies, Coralville, IA, USA, cat. #174079688, 174079689), serving as the endogenous control and for normalization (results were normalized with the control column).

### 2.4. Primary Human Glioblastoma Cells

Part of the sample was enzymatically digested with collagenase/hyaluronidase (StemCell Technologies, Vancouver, BC, Canada, cat. # 07912) at 37 °C for 1 h in RPMI medium (Sigma-Aldrich, cat. #R8758) with rocking. Tumor suspensions were filtered through 70 μM sterile nylon gauze and mixed with Dulbecco’s modified Eagle’s medium (DMEM, Sigma-Aldrich, cat. #D7777), supplemented with 10% FBS, 50 U/mL penicillin, and 50 µG/mL streptomycin and maintained in a humidified atmosphere of 5% CO_2_ and 95% air at 37 °C. The glioblastoma cell lines used in this study had been cultivated for no more than 16 passages.

### 2.5. Preparation of Microglia-Conditioned Medium

HMC3 microglia (ATCC, Rockefeller, VA, USA) and glioma cells were seeded in co-culture in T-75 flasks at a 1:1 ratio in DMEM for 24 h. Co-cultured cells were incubated in serum-free medium for 24 h prior to obtaining microglia-conditioned medium (MCM). MCM was centrifuged to remove debris and dead cells and immediately used for experiments.

### 2.6. Western Blot Analysis

Glioma cells were plated in 6-well plates and incubated under normal conditions for 24 h, and the medium was replaced with serum-free medium 24 h prior to experiments. Cells were divided into the following groups: control group, in which cells were incubated in serum-free medium; MCM treatment; treatment with one of the following cytokines for 30 min: 5 μM platelet-derived growth factor α (PDGFα, StemCell Technologies, Tukwila, WA, USA, cat. #78095), 5 μM PDGFβ (StemCell Technologies, cat. #78097), 100 μM stromal-derived factor 1 α (SDF-1α, Bio-Rad, cat. #PHP1222), 100 μM IL-6 (StemCell Technologies, cat. #78050), 10 μM IL-8 (StemCell Technologies, cat. #78084), or 1 μM EGF (StemCell Technologies, cat. # 78006.1). In the inhibitor group, cells were pre-treated for 30 min with their respective receptor inhibitor: 50 nM PDGFR tyrosine kinase inhibitor III, 200 nM DMPQ (5,7-Dimethoxy-3-(4-pyridinyl)quinoline) dihydrochloride (Santa Cruz Biotechnology, Dallas, TX, USA, cat. #203927 and #204173, respectively), 1 μM burixafor hydrobromide, 100 ng/mL tocilizumab, 1 μM reparixin (MedChemExpress, Monmouth Junction, NJ, USA, cat. #19867A, #P9917, and #15251, respectively), and 1 μM gefitinib (Sigma-Aldrich, cat. #SML1657), followed by the corresponding treatment in combination with MCM for 120 min. Next, the cells were lysed in RIPA buffer, and immunoblotting was performed as described previously [30]. Briefly, protein concentrations were determined using a Bradford assay kit (Bio-Rad, cat. #5000006). Ten microgram protein samples were resolved with 10% SDS-PAGE and transferred onto Amersham Hybond ECL nitrocellulose membranes (GE Healthcare Bio-Sciences Corp., Pittsburgh, PA, USA). The antibodies used in this study were developed against p-Pyk2 (Y579/Y580) (1:500, Invitrogen, Carlsbad, CA, USA, cat. # 44-636G), Pyk2 (1:1000), p-FAK (Y925) (1:1000), FAK (1:1000) (Cell Signaling Technology, Danvers, MA, USA, cat. #3292, #3284, #3285, respectively). Total protein staining with REVERT Total Protein Stain (LI-COR Biotechnology, cat. # 926-11016) was used as a loading control [31]. The signals were visualized using the Odyssey CLx Quantitative Infrared Imaging System with the secondary infrared antibodies IRDye 800CW goat anti-rabbit (1:25,000, LI-COR Biotechnology, cat. #925-32211). Results were analyzed with Image Studio Lite Software, version 5.2 (LI-COR Biotechnology, Lincoln, NE, USA).

### 2.7. Cell Viability

Cell viability was determined by performing a trypan blue exclusion assay. Primary human glioblastoma cells were plated in 60 mm × 15 mm petri dishes in serum-free medium at 250,000 cells per dish and incubated for 3 days with and without MCM, the cytokines PDGFα, PDGFβ, SDF-1α, IL-6, IL-8, and EGF, and with MCM in combination with the respective cytokine receptor inhibitors, PDGFR tyrosine kinase inhibitor III, DMPQ dihydrochloride, burixafor, tocilizumab, reparixin, and gefitinib, in the concentrations indicated above. The cells were then harvested and incubated with trypan blue. The total number of live cells (trypan blue negative) was determined by cell counting.

### 2.8. RNA Interference Using Small Double-Stranded RNAs

Glioma cells were transfected with siRNA targeting Pyk2 (Qiagen Inc., Valencia, CA, USA, cat. #SI02649031, #100159039) and FAK (Qiagen Inc., cat. #S100301532, #103103156) using Lipofectamine RNAiMAX reagent (Invitrogen, cat. #13778-150) according to the manufacturer’s instructions. Cells (5.0 × 10^4^) were plated in 6-well plates and transfected with 20 nM Pyk2 siRNA or 20 nM FAK siRNA using 3 µL of Lipofectamine RNAiMAX in serum-free medium. Stealth RNAi negative control duplex (Invitrogen, cat. #12935) was used as control. The efficacy of transfection was controlled by Block-iT Alexa Fluor red Fluorescent Oligo (Invitrogen, cat. #14750) and by RT-PCR. Forty-eight hours after transfection, the cells were used for cell cycle, migration, and invadopodia assays.

### 2.9. Invadopodia Assay

The cells were transfected with siRNA against Pyk2 or FAK and plated on fluorescein-conjugated, gelatin-coated glass coverslips (gelatin concentration 0.2 mg/mL, Invitrogen, cat. #G13187) for 16–48 h (the optimum incubation time was determined for each cell culture). Next, the cells were fixed with 4% paraformaldehyde for 10 min at 37 °C, permeabilized with 0.5% Triton-X 100 for 15 min, and blocked in 1% BSA solution for 1 h before incubation with phalloidin–tetramethyl–rhodamine (diluted 1:500, Sigma-Aldrich, cat. #P1951), and stained with DAPI (Sigma-Aldrich, cat. #D9542). Fluorescent images were visualized using an Olympus Fluoview FV1000 confocal microscope (Olympus Corporation) with 40× oil immersion objectives. Images were analyzed using Image J software (http://imagej.nih.gov/ij, version 1.52a, assessed on 1 October 2021). For the quantification of invadopodia formation, the cells forming invadopodia in each digital image were counted and normalized to the number of total cells in the same image. To quantify invadopodia activity, black and white images of gelatin degradation were analyzed, and the percentage of matrix degraded was normalized to the number of nuclei in each image, as measured from DAPI staining.

### 2.10. Migration Assay

Migration assays were performed using Fluoroblok inserts (8 µM pore size, VWR Scientific, Batavia, IL, USA). Serum-starved cells (30,000) were placed on the insert membrane, and the assays were performed following the addition of PDGFα (5 μM), PDGFβ (5 μM), SDF-1α (100 μM), IL-6 (100 μM), IL-8 (10 μM), EGF (1 μM), or MCM in combination with the corresponding receptor inhibitors, PDGFR tyrosine kinase inhibitor III (50 nM), DMPQ dihydrochloride (200 nM), burixafor (1 μM), tocilizumab (100 ng/mL), reparixin (1 μM), or gefitinib (1 μM) to the lower compartment. After 8–23 h (the optimum migration time was determined for each cell line), the cells were fixed with ice-cold 70% ethanol and stained with propidium iodide. The number of cells that had migrated to the lower compartment was determined by counting the number of fluorescent cells.

### 2.11. Cell Cycle Assay

The cell cycle phase was evaluated using flow cytometry analysis. The Guava^®^ Cell Cycle Reagent Kit (Luminex Corporation, Hayward, CA, USA, cat. #4500-0220), containing propidium iodide, was used to determine cell cycle phase distribution. Glioma cells were transfected with siRNA against FAK and Pyk2 and treated with PDFGα (5 μM), PDGFβ (5 μM), SDF-1α (100 μM), IL-6 (100 μM), IL-8 (10 μM), or EGF (1 μM), as well as with MCM in the presence of the corresponding receptor inhibitors: PDGFR tyrosine kinase inhibitor III (50 nM), DMPQ dihydrochloride (200 nM), burixafor (1 μM), tocilizumab (100 ng/mL), reparixin (1 μM), or gefitinib (1 μM), respectively. Next, the cells were washed with PBS, fixed with ice-cold 70% ethanol for 24 h, resuspended in propidium iodide-containing reagent for 30 min at room temperature, and analyzed with the Guava easyCyte flow cytometer (Luminex Corporation) and the InCyte software module (Luminex Corporation). Each experiment was performed in triplicate.

### 2.12. Statistical Analysis

Data are presented as mean ± standard deviation (SD). Statistical comparisons were performed with one-way analysis of variance (ANOVA). A *p*-value of <0.05 was considered significant. When a significant overall effect was present, intergroup comparisons were performed using a Tukey–Kramer correction for multiple comparisons in GraphPad Prism 9.1.0 statistical software (San Diego, CA, USA).

## 3. Results

### 3.1. Correlation between Gene Expression of Cytokines in Microglia and Pyk2 and FAK Protein Expression in Glioma Cells from GBM Tumor Samples

Recent studies [10,25,27] suggest that microglial-derived factors drive the expression of Pyk2 and FAK in glioma cells to promote their proliferation and invasiveness. To determine the association between the gene expression of microglial cytokines/chemokines and Pyk2 and FAK gene and protein expression in glioma cells, a Pearson’s correlation test was performed. Using RT-PCR, the Percoll-purified microglia fraction from 20 human GBM specimens was analyzed for the gene expression of key cytokines and chemokines (encoding EGF, PDGFα, PDGFβ, NGF, IL-6, IL-8, IL-10, CXCR12, and CCL5). The glioma cell tumor fraction was analyzed for the level of gene expression of corresponding receptors, as well as Pyk2 and FAK. The protein expression of Pyk2 and FAK and the phosphorylation levels of Pyk2 (Y579/Y580) and FAK (Y925) were evaluated by western blotting. Pierson correlation analysis was performed, and cytokine pairs with a strong correlation (0.6–1.0) were selected for further study (Figure 1).

A strong positive correlation (r > 0.6) between Pyk2 and FAK gene expression and cytokine/chemokine receptors was observed in glioma fractions. Statistically significant correlations of gene expression of Pyk2 with that of EGFR (r = 0.93, *p* < 0.05), PDGFβR (r = 0.99, *p* < 0.05), NGFR (r = 0.91, *p* < 0.05), CXCR1 (also known as the IL-8 receptor, r = 0.68, *p* < 0.05), and CXCR4 (also known as SDF-1αR, r = 0.6, *p* < 0.05) were observed. Similar results were obtained when comparing FAK with EGFR and PDGFβR (r = 0.71 and r = 0.63, respectively, *p* < 0.05). Despite the fact that no significant correlation between Pyk2 and FAK gene expression in glioma and the corresponding cytokines in microglia was found, a strong positive correlation for phosphorylated and total Pyk2 and FAK protein expression in glioma cell fractions and cytokine expression in microglia was observed. Phosphorylated and total Pyk2 protein expression demonstrated a strong correlation with EGF (r = 0.8, *p* < 0.05), PDGFα (r ≥ 0.81, *p* < 0.05), PDFGβ (r > 0.92, *p* < 0.05), IL-8 (r > 0.78, *p* < 0.05), and SDF-1α (r > 0.93, *p* < 0.05). Similar results were obtained when FAK was evaluated. Of note, no correlation was found between listed microglial cytokines/chemokines and glioma Pyk2 and FAK intensity of phosphorylation, calculated as a proportion of phosphorylated protein to total protein expression. This result might indicate that the observed correlation between microglial factors and the level of phosphorylated Pyk2 and FAK in glioma cells is due to an impact on protein expression, but not phosphorylation. By contrast, IL-6 expression in microglia did not demonstrate a correlation with Pyk2/FAK phosphorylated and total protein expression; however, it was correlated with the intensity of Pyk2 phosphorylation (Pyk2 (Y579/Y580))/total Pyk2), indicating the possible regulation of Pyk2 phosphorylation by this cytokine.

No correlation was observed between the genes encoding Pyk2 and FAK and their phosphorylated and total protein expression. Taken together with the lack of correlation between expression of the genes encoding Pyk2 and FAK and microglial factors, this finding suggests a microglial-dependent regulation of Pyk2 and FAK signaling in glioma only at the protein expression level. In addition, cytokine/chemokine gene expression in microglia and their respective receptors in glioma were not significantly correlated.

Based on these results, EGF, PDGFα, PDFGβ, IL-6, IL-8, and SDF-1α were selected as candidates for the microglial regulation of Pyk2 and FAK signaling in glioma cells. The effects of these factors on Pyk2- and FAK-related glioma cell proliferation and dispersal were further investigated in three primary human GBM cell lines developed from the analyzed specimens. The three selected cell lines represent several basal levels of Pyk2 and FAK expression. The patient information and cell line characterizations are provided in Appendix A. Cell line 1 (CL1) is characterized by high expression of total Pyk2 (Appendix A) and a low expression level of FAK (Appendix A) with moderate phosphorylation of Pyk2 (Appendix A) and high phosphorylation of FAK (Appendix A). Cell line 2 (CL2) is characterized by low total expression of Pyk2 (Appendix A) with moderate phosphorylation (Appendix A) and high total FAK expression (Appendix A) with low phosphorylation (Appendix A). Cell line 3 (CL3) is characterized by moderate total Pyk2 expression (Appendix A) with moderate phosphorylation (Appendix A) and high total FAK expression (Appendix A) with low phosphorylation (Appendix A).

### 3.2. Cytokines Released by Microglia Activate Pyk2 and FAK Signaling Pathways in Human Glioma Cells

To identify the microglial factors activating Pyk2 and FAK signaling in glioma cells, the phosphorylated (p-Pyk2 (Y579/Y580) and p-FAK (Y925)) and total protein levels of these kinases were evaluated upon cell treatment with MCM or with each cytokine/chemokine identified above (EGF, PDGFα, PDGFβ, SDF-1α, IL-6, and IL-8). Additionally, the effect of treatment with MCM in combination with any of the inhibitors for cytokine/chemokine receptors was evaluated. Because of inter-individual variability, we chose to present the results of these experiments for each cell line individually.

CL1: Figure 2a shows that SDF-1α, IL-8, IL-6, and EGF significantly upregulated Pyk2 phosphorylation compared with control (≥29% for all, *p* < 0.05). From this group, IL-6 induced the strongest activation of Pyk2 phosphorylation (260%, *p* < 0.05). The total protein levels of Pyk2 (Figure 2b) were not significantly increased, except for SDF-1α treatment (47%, *p* < 0.05), indicating that the observed increase in phosphorylated Pyk2 in the presence of SDF-1α is due to the increase in total Pyk2 protein and not to increased phosphorylation. EGF also appears to promote an increase in total protein levels of Pyk2, although this effect did not reach statistical significance (38%, *p* = 0.07). MCM did not affect the activation of Pyk2 in CL1. Correspondingly, blockers of SDF-1α, IL-6, IL-8, and EGF receptors (burixafor, tocilizumab, reparixin, and gefitinib, respectively) in the presence of MCM did not affect Pyk2 phosphorylation. A possible explanation of this effect is the extremely high autocrine gene expression of SDF-1α, IL-8, and EGF by CL1 glioma cells (Appendix A) and, consequently, a high basal level of Pyk2 expression and phosphorylation in this cell line (Appendix A). Therefore, microglial-derived cytokines/chemokines did not provide an additive effect for the activation of Pyk2, which was basally supported by autocrine mechanisms. For this reason, cells treated with chemokine/cytokine receptor blockers individually did not significantly affect the activation of Pyk2 due to the strong stimulatory effect of other cytokines released by glioma cells.

In contrast to the insensitivity of p-Pyk2 to MCM, FAK was activated by MCM by 31% compared with control (*p* < 0.05, Figure 2c). Additionally, all cytokines evaluated, except for IL-8, enhanced the phosphorylation of FAK (by ≥38% fold, *p* < 0.05 for each). Treatments with inhibitor III, burixafor, and gefitinib reduced the stimulatory effect of MCM (by between 21 and 69%, *p* < 0.05 for each), supporting the roles of PDGFα, SDF-1α, and EGF in the MCM-mediated activation of FAK. Despite the stimulatory effect of PDGFβ and IL-6 on FAK phosphorylation, their corresponding inhibitors, DMPQ and tocilizumab, did not reverse MCM-dependent phosphorylation of FAK in this cell line, probably due to the strong stimulatory effect of other chemokines, such as SDF-1α and EGF, that are present at high concentrations in MCM (a cytokine expression profile of microglia treated with medium conditioned from each of the investigated cell lines is provided in Appendix A). Changes in the total expression of FAK were observed with SDF-1α and EGF (Figure 2d, 37 and 60% increase, respectively, *p* < 0.05 for each), which were the same as those observed for the total expression of Pyk2 protein (Figure 2b). Of note, treatment with burixafor, tocilizumab, reparixin, and gefitinib increased total FAK in the presence of MCM by between 55 and 81% (*p* < 0.05, for each). These results suggest roles for SDF-1α, IL-6, and EGF in upregulation of both Pyk2 and FAK phosphorylated proteins, while PDGFα and PDGFβ are involved only in FAK regulation when it is related to CL1.

CL2: Figure 2e shows that MCM increased phosphorylated Pyk2 by 19% compared with control (*p* < 0.05). Of all the cytokines and chemokines evaluated, only EGF significantly upregulated Pyk2 phosphorylation (by 39%, *p* < 0.05), despite non-significant upregulation of p-Pyk (Y579/Y580), which was also observed for SDF-1α, IL-6, and IL-8. However, inhibitor III, DMPQ, burixafor, reparixin, and gefitinib inhibited the activation of Pyk2 in the presence of MCM by between 23 and 46% (*p* < 0.05). This non-significant upregulation of Pyk2 phosphorylation by SDF-1α, IL-6, and IL-8, together with the strong inhibitory effect of the corresponding receptors’ blockers, can be explained by the high autocrine expression of SDF-1α, IL-6, and IL-8 by CL2 (Appendix A). The lack of stimulatory effect of PDGFα and PDGFβ on Pyk2 phosphorylation, together with the inhibitory effect of their corresponding blockers on the MCM-dependent upregulation of p-Pyk2 (Y579/Y580), can be explained by the complex recognition between PDGFs and PDGFRs and the multitude of possible PDGF–PDGFR interactions. PDGFRα are activated by homodimeric PDGF-αα, PDGF-ββ, and heterodimeric PDGF-αβ. Additionally, the heterodimerization of PDGFRαβ can be induced by the PDGF-ββ homodimer or the PDGF-αβ heterodimer [32]. Taking in account this complexity, the application of single PDGFα or PDGFβ may not result in the activation of Pyk2. However, inhibition of either PDGFRα or PDGFRβ may affect PDGFR dimerization and signaling. Deep studies directed at the elucidation of the mechanisms of PDGFR signaling on the activation of Pyk2 are needed. In a frame of the current study, our findings support the involvement of PDFGα, PDGFβ, SDF-1α, IL-6, IL8, and EGF in the activation of Pyk2 by MCM. Moreover, SDF-1α, IL-6, and EGF promoted the elevation of the expression of total Pyk2 (Figure 2f) to a similar extent (by between 44 and 48%, *p* < 0.05).

Figure 2g demonstrates that MCM also upregulated FAK phosphorylation in CL2 compared with control (by 34%, *p* < 0.05). Similarly, PDFGβ, IL-6, and EGF increased p-FAK (Y925) expression by 69%, 58%, and 285%, respectively (*p* < 0.05). Inhibitor III, DMPQ, and gefitinib partially reversed the effect of MCM on FAK phosphorylation by between 35% and 46% (*p* < 0.05), while an inhibitory effect of the IL-6R blocker tocilizumab was not observed. Taking into account the high levels of expression of PDGFβ and EGF, together with the low expression of IL-6 in microglia/CL2-conditioned medium (Appendix A), we propose that an inhibitory effect of tocilizumab on FAK phosphorylation was masked by a strong activation effect of PDGFβ and EGF in MCM. These results suggest that FAK is activated by microglia-derived PDGFβ, IL-6, and EGF in CL2.

Figure 2h indicates an increase in total FAK expression induced by IL-6 and EGF compared with control (by 19% and 84%, respectively, *p* < 0.05). Treatment with inhibitor III, DMPQ, or burixafor decreased total expression of the kinase (by 33% for inhibitor III or DMPQ and by 15% for burixafor, *p* < 0.05), which is consistent with FAK phosphorylated protein modifications in MCM-treated cells. These findings support the role for EGF, PDGFβ, SDF-1α, IL6, and IL-8 in the activation of Pyk2 and FAK signaling in CL2.

CL3: Figure 2i illustrates that, as with CL2, p-Pyk2 (Y579/Y580) in CL3 was significantly upregulated by MCM compared with control (by 37%, *p* < 0.05). Consistent with the results in CL1, activation of p-Pyk2 (Y579/Y580) was observed in the presence of IL-6 (217%), IL-8 (81%), and EGF (68%, *p* < 0.05 for each). The MCM-dependent increase in phosphorylated Pyk2 was significantly decreased by DMPQ, burixafor, reparixin, and gefitinib (≥40% for all, *p* < 0.05), supporting the involvement of PDGFβ and SDF-1α in the activation of Pyk2, in addition to IL-6, IL-8, and EGF. The lack of inhibitory effect of the IL-6R blocker tocilizumab in the presence of MCM, together with the strong Pyk2 (Y579/Y580) upregulation in response to IL-6, can be explained by the high expression level of EGF and IL-8 in microglia/CL3 conditioned medium (Appendix A). The lack of effect of PDGFβ and SDF-1α on Pyk2 phosphorylation, together with the inhibitory effect of their corresponding receptors’ blockers, DMPQ and burixafor, might be due to the low expression level of PDGFRβ and CXCR12, together with their autocrine activation, which may exceed the receptor activation capacity (Appendix A). However, significant downregulation of Pyk2 phosphorylation by DMPQ and burixafor indicates the important role of PDGFβ and SDF-1α in the regulation of Pyk2 signaling.

In contrast to CL1 and CL2, a significant increase in total Pyk2 was observed in MCM-treated cells compared with control (by 62%, *p* < 0.05, Figure 2j). Both IL-6 and EGF enhanced this parameter by 23% and 34%, respectively (*p* < 0.05). Treatment with burixafor and gefitinib reduced the total expression of Pyk2 in MCM-treated cells (by 39% and 55%, respectively, *p* < 0.05).

Treatment with MCM also upregulated FAK phosphorylation by 30% (*p* < 0.05, Figure 2k). Similar to observations in CL1 and CL2, IL-6 and EGF activated FAK by 32% and 73%, respectively (*p* < 0.05). Of note, the activating effect of MCM was reversed by inhibitor III, DMPQ, burixafor, reparixin, and gefitinib (by ≥21% for all, *p* < 0.05), suggesting a role for PDGFα, PDGFβ, SDF-1α, and IL-8 in the activation of FAK signaling. Total FAK expression was increased by EGF by 72% compared with control (Figure 2l). No effect of MCM on this parameter was observed. However, reparixin and gefitinib reversed the effect of MCM on total FAK expression (by 50% for both, *p* < 0.05). 

Original uncropped western blot images and loading controls with REVERT Total Protein Stain are presented in Appendix A.

These results suggest that the activation of Pyk2 and FAK signaling is mostly IL-6- and EGF-dependent in CL3, while PDGFβ, SDF-1α, and IL-8 also contribute to Pyk2 and FAK phosphorylation. Altogether, these findings indicate that, in all three evaluated specimens, Pyk2 and FAK signaling pathway activation significantly depend on microglia-derived EGF, PDGFβ, SDF-1α, IL-6, and IL-8, although the extent of their regulatory contribution differs from patient to patient.

### 3.3. Cytokines/Chemokines Released by Microglia Promote Glioma Cell Extracellular Matrix Degradation through Pyk2 and FAK Signaling

Glioma cell invasion is a complex process composed of extracellular matrix degradation and migration of the cell. As different intracellular and extracellular signaling pathways can be involved, in this study, the regulation of matrix degradation and migration were investigated independently.

Invadopodia are actin-rich membrane protrusions that serve as mediators of cell invasion [33]. To evaluate the role of the cytokines and chemokines investigated in the Pyk2- and FAK-dependent formation of functional invadopodia and extracellular matrix degradation, a FITC-conjugated gelatin-degradation assay was performed. The number of cells that form invadopodia (invadopodia formation, IF) and the portion of gelatin matrix degradation (invadopodia activity, IA) were evaluated for the CL1, CL2, and CL3 cell lines. siRNA knockdown against Pyk2 or FAK, together with mock control cells, were treated with MCM or any one of the cytokines investigated (EGF, IL-6, IL-8, SDF-1α, PDGFα, or PDGFβ), as well as with MCM in combination with the corresponding inhibitors of cytokine receptors (gefitinib, tocilizumab, reparixin, burixafor, inhibitor III, or DMPQ). The efficacy of Pyk2 and FAK knockdown is presented in Appendix A).

CL1: Glioma cells from CL1 did not exhibit IF and extracellular matrix degradation under control conditions (Figure 3c,d). However, treatment with MCM induced both IF and IA, with 73% of the cells forming invadopodia and 0.15% degradation of extracellular matrix (*p* < 0.05). siPyk2 and siFAK knockdown abolished this effect of MCM (Figure 3b–d). These results demonstrate a stimulatory effect of MCM on functional invadopodia formation in glioma cells that is Pyk2 and FAK dependent.

The results presented in Figure 3c demonstrate that PDGFβ, SDF-1α, and IL-6 increased IF in the mock control experimental group to a similar extent as MCM treatment (70%, 83%, and 91% of cells formed invadopodia, respectively, *p* < 0.05, Figure 3b), and there was an increase in IA to 0.32%, 0.35%, and 0.2% of degraded matrix, respectively, compared with the complete lack of matrix degradation in control (Figure 3d). IL-8 slightly increased both parameters (by 12% IF and by 0.05% IA), although it did not reach statistical significance (*p* = 0.37). The stimulatory effect of PDGFβ, SDF-1α, and IL-6 on IF and IA was reversed by either siPyk2 or siFAK knockdown. Inhibitors of the corresponding receptors, DMPQ, burixafor, tocilizumab, and reparixin, eliminated the stimulatory effect of MCM on IF and IA (Figure 3e,f), supporting the involvement of PDGFβ, SDF-1α, IL-6, and IL8 in MCM-induced CL1 glioma cell invasion. These findings suggest that Pyk2 and FAK are key regulators of IF and IA in CL1 through PDGFβ, SDF-1α, and IL-6.

CL2: In contrast to CL1, CL2 presented strong IF and AI under basal conditions, which are FAK dependent (Figure 3g,i,j). Indeed, siFAK knockdown reduced the IF by 39% (*p* < 0.05) compared with the control group, together with a 60% reduction in AI. Despite the fact that Pyk2 knockdown did not significantly affect IF and IA compared with the control, it increased stress-fiber tension, inducing cells to pull and tear the extracellular matrix (Figure 3g). The total number of cells with invadopodia was not affected by MCM (Figure 3h,i); however, AI resulted in an increase in extracellular matrix degradation by 430% (*p* < 0.05) compared with the control group in a manner that is Pyk2- and FAK-dependent (a 94% decrease, *p* < 0.05, Figure 3j). Similarly, siPyk2 and siFAK reduced IF by 47 and 61%, respectively, following MCM treatment (*p* < 0.05). Of note, siRNA knockdown against Pyk2 increased the tension of cell attachment to the extracellular matrix in MCM treatment in the same way as in the control group, although to a lesser extent. Therefore, Pyk2 could be involved in the regulation of cell detachment, which is important for the cell migration process. In addition, the reduced number of cells forming invadopodia after siPyk2 and siFAK knockdown indicates the relevance of Pyk2 and FAK in MCM-induced glioma invasion.

Figure 3i demonstrates that none of the cytokines evaluated in CL2 significantly affected IF compared with controls. However, PDGFβ and SDF-1α increased the IA by 4.7- and 2.5-fold, respectively, compared with control (*p* < 0.05, Figure 3f). This stimulatory effect was abolished by siPyk2 and siFAK (by ≥87% for both kinases, *p* < 0.05), indicating that they are parts of the PDGFβ and SDF-α signaling pathways controlling the activity of invadopodia. No significant changes were observed in IF and in AI in the presence of EGF, PDGFα, IL-6, and IL-8. For these treatments, siPyk2 knockdown resulted in increased tension to the matrix, the same as observed in the siPyk2 control group, indicating that these factors are not involved in the regulation of invadopodia in CL2.

Interesting results were obtained from cell treatment with MCM in combination with cytokine/chemokine blockers (Figure 3k,l). While an increase in AI was detected for PDGFβ and SDF-1α treatments only (Figure 3j), the MCM effect on IA was abrogated by their corresponding receptor blockers, DMPQ and burixafor, and also by inhibitors of EGFR, IL-6R, and IL-8R (gefitinib, tocilizumab, and reparixin, respectively; by ≥50% for all, *p* < 0.05; Figure 3l), indicating the autocrine regulation of IF and IA. These findings suggest that both Pyk2 and FAK signaling, mediated by PDGFβ and SDF-1α, are involved in the regulation of IF and IA (with possible autocrine regulation by EGF, IL-6, and IL-8) in CL2. In addition, the regulation of extracellular matrix detachment could also be mediated by Pyk2.

CL3: IF and AI under basal conditions in CL3 were similar to what was observed for CL2 (Figure 3m,o,p). In contrast to CL2, silencing of FAK under control conditions increased both IF by 69% (*p* < 0.05) and AI (by 280%, *p* < 0.05), suggesting an anti-matrix degradation role of this kinase in CL3. Pyk2 knockdown led to increased matrix attachment and stress-fiber tension, resulting in folding and tearing of the matrix, similar to CL2 (Figure 3m). As shown in CL1, treatment with MCM increased IF and IA (by 78% and 83%, respectively, *p* < 0.05) in CL3, and this activation was reversed by Pyk2 knockdown, but not by FAK knockdown (Figure 3n–p). Indeed, siPyk2 decreased the stimulatory effect of MCM on IF by 4-fold, and IA by 10-fold (*p* < 0.05). These findings indicate that in CL3 both Pyk2 and FAK are involved in the regulation of extracellular matrix degradation, with Pyk2 acting as a booster of matrix degradation and cell mobility, and FAK reducing the activity of matrix degradation.

Figure 3o demonstrates that EGF, PDGFβ, SDF-1α, and IL-8 increased the number of cells with invadopodia (by ≥170% for all, *p* < 0.05). A stimulatory effect of EGF, PDGFβ, and SDF-1α was also observed in IA (by 890%, 370%, and 400%, respectively, *p* < 0.05 for each; Figure 3p). This effect was inhibited by siPyk2 (by ≥ 55% for all, *p* < 0.05), indicating that Pyk2 is part of the EGF, PDGFβ, and SDF-1α signaling pathways in the promotion of cell invasion. Additionally, the inhibitors of EGFR, PDGFβR, and SDF-1αR (gefitinib, DMPQ, and burixafor) and, to a lesser extent, the inhibitor of IL-8R (reparixin), significantly reduced FI and AI in the presence of MCM (by ≥300% for gefitinib, DMPQ, and burixafor, and by 30% for reparixin, *p* < 0.05; Figure 3q,r), supporting an important role for microglia-derived EGF, PDGFβ, SDF-1α, and IL-8 in extracellular matrix degradation in CL3. By contrast, IL-6 reduced IF by 73% compared with control (*p* < 0.05). IL-6 in the presence of Pyk2 knockdown resulted in increased damage to the extracellular matrix caused by stress-fiber tension. This observed effect does not allow us to determine whether Pyk2 signaling in CL3 is mediated by IL-6. However, considering the lack of effect by tocilizumab in MCM-stimulated IF and IA, the role of IL-6 in microglia-driven extracellular matrix degradation in CL3 is negligible.

Altogether, these findings suggest that in the three evaluated cell lines, both Pyk2 and FAK signaling are involved in the regulation of extracellular matrix degradation, with microglia-derived PDGFβ and SDF-1α being common upstream regulators of Pyk2 and FAK for all three cell lines. EGF, IL-6, and IL-8 are also involved in Pyk2- and FAK-dependent extracellular matrix degradation; however, the extent of their involvement is patient dependent.

### 3.4. Cytokines/Chemokines Released by Microglia Promote Glioma Cell Migration through Pyk2 and FAK Signaling

Transwell migration assays were performed for all three cell lines investigated in order to evaluate the role of Pyk2 and FAK, as well as to identify their extracellular upstream regulators, which lead to increased migration capacity of glioma cells. Mock cells and cells that were siRNA-transfected against Pyk2 or FAK were used for the assay with PDGFα, PDGFβ, SDF1α, EGF, IL6, IL8, or MCM or MCM in combination with corresponding chemokine/cytokine receptor inhibitors in the lower compartment. The migration times were found to be 23 h for CL1 and 8 h for both CL2 and CL3.

A significant reduction in migration in Pyk2- and FAK-knockdown cells under basal conditions was detected in CL1 and CL3 (Figure 4a,e). A reduction in migration was also observed in CL2; however, this did not reach statistical significance. Of note, in all the investigated cell lines, stimulation with PDGFβ or EGF drastically increased migration, and this effect was eliminated by either Pyk2 or FAK knockdown (Figure 4a,c,e). In addition, PDGFα induced the Pyk2- and FAK-dependent activation of migration in CL2 and CL3 (Figure 4c,e), while SDF-1α induced the Pyk2- and FAK-dependent activation of migration in CL1 and CL3 (Figure 4a,e).

MCM resulted in 30–300% upregulation of migration in all cell lines (Figure 4b,d,f; *p* < 0.05). Knockout of either Pyk2 or FAK reduced this effect. Inhibitors of PDGFRβ, EGFR, or SDF-1αR (DMPQ, gefitinib, and burixafor, respectively) significantly reduced the stimulatory effect of MCM in all cell lines, confirming the key role of microglia-derived PDGFβ, EGF, and SDF-1α in glioma cell migration (Figure 4b,d,f). No additive effects of Pyk2 and FAK knockdown in DMPQ treatment were detected in any of the cell lines, indicating that PDGFβ stimulates migration through both Pyk2 and FAK. The additive effect of Pyk2 and FAK knockdown in gefitinib and burixafor treatment was detected in CL1 (Figure 4b), but not in other cell lines, indicating that pathways other than the Pyk2 and FAK pathways may be involved in EGF- and SDF-1α-mediated activation of migration in CL1. The lack of effect of SDF-1α in CL2, together with the strong inhibitory effect of burixafor (Figure 4d), could be a consequence of the significant autocrine expression of SDF-1α in CL2, which was confirmed by PCR analysis (Appendix A). Therefore, SDF1α, together with EGF and PDGFβ, can be classified as a key microglia-derived stimulator of migration in all cell lines.

Despite the fact that PDGFα significantly stimulated migration in CL2 and CL3 (Figure 4a,e), inhibitor III (blocker of PDGFRα) did not reduce the effect of MCM in any of the investigated cell lines (Figure 4b,d,f). RT-PCR analysis of cytokine/chemokine expression in microglia pre-activated with medium conditioned from glioma cells, did not detect expression of PDGFα in microglia pre-activated by CL2 and CL3 but detected trace expression of PDGFα in microglia pre-activated by CL1 (Appendix A). PDGFα gene expression was also not detected in any of the glioma cell lines investigated (Appendix A). These results suggest that, despite the findings that PDGFα can be actively involved in the regulation of glioma cell migration and that this regulation is carried out through Pyk2 and FAK signaling, the absence or low expression of PDGFα in the glioma–microglia microenvironment reduces the role of this factor in microglia-driven glioma cell migration.

### 3.5. Microglial-Derived EGF, SDF1α, and IL-8 Stimulate Glioma Cell Viability

To determine the role of microglia-derived factors on glioma cell viability, the trypan blue exclusion test was used. Cells were incubated for 72 h with the following treatments: control, PDGFα, PDGFβ, EGF, SDF-1α, IL-6, IL-8, MCM, and MCM in combination with inhibitors of the corresponding receptors.

A significant increase in the number of viable cells after 72 h of incubation with MCM compared with control was observed in all investigated cell lines (by 74% in CL1, 37% in CL2, and 64% in CL3, *p* < 0.05 for each, Figure 5). Incubation with EGF or IL-8 resulted in a significant increase in the number of cells at 72 h compared with control in all cell lines (by 37–78%, *p* < 0.05). The EGFR and IL-8R blockers, gefitinib and reparixin, respectively, abolished the MCM-enhanced viability effect, reducing the number of viable cells under each treatment in all cell lines (by 19–41%, *p* < 0.05).

SDF1α increased viability in CL1 and CL3 (by 68 and 62%, respectively, *p* < 0.05; Figure 5a,c), while burixafor reduced the stimulatory effect of MCM in all investigated cell lines. Considering the strong autocrine gene expression of SDF-1α in CL2 (Appendix A), the absence of an effect of SDF1α on cell viability in this cell line could be due to the saturation of CXCR4 (also known as SDF-1αR) receptor activation. Therefore, SDF1α, together with EGF and IL-8, are key regulators of cell viability in the glioma–microglia microenvironment, in all cell lines.

IL-6 upregulated cell viability in CL1 and CL2. This result is correlated with the reduction in the MCM effect by the inhibitor of IL-6R, tocilizumab, in CL1 and CL2 (by 16 and 13%, respectively, *p* < 0.05) but not in CL3, indicating that cell viability in CL3 is IL-6-independent.

PDGFβ significantly increased cell viability only in CL2, promoting a 2.1-fold increase (*p* < 0.05). In CL1 and CL3, a non-significant increase in cell viability was observed under PDGFβ stimulation. Correspondingly, DMPQ significantly reduced the effect of MCM in CL2 only (a 16% decrease, *p* < 0.05) along with a non-significant reduction in CL1 and CL3, suggesting that PDGFβ could be involved in microglia-driven glioma cell viability. However, the extent of this regulation is patient dependent.

### 3.6. Microglia-Derived EGF and IL-6 Induce Mitosis through Pyk2 and FAK Signaling

In order to further investigate the role of Pyk2 and FAK in microglia-driven glioma cell proliferation and to identify the upstream signaling mechanisms, cell cycle analysis was performed for mock control and siRNA knockdown against Pyk2 and FAK cells. Cells were incubated for 24 h with the following treatments: control, PDGFα, PDGFβ, EGF, SDF-1α, IL-6, IL-8, MCM, and MCM in combination with each of the inhibitors of the corresponding receptors. The cell distribution at defined cell cycle phases was analyzed by flow cytometry.

Cell cycle analysis of glioma cells with knockdown of Pyk2 or FAK revealed that the involvement of Pyk2 and FAK in cell cycle regulation is patient dependent. In CL1, FAK knockdown resulted in significant accumulation of cells in G2/M (35%, *p* < 0.05) and a strong reduction in cells in the S phase (35%, *p* < 0.05; Figure 6a), indicating cell cycle arrest at G2/M. In CL2, FAK silencing decreased the percentage of cells in the G2/M phase by 12% (*p* < 0.05) and increased the accumulation of cells in G1 (3%, *n*.s. *p* = 0.58), indicating that, in CL1 and CL2, FAK is involved in the regulation of cell cycle progression and stimulates cell proliferation. The dramatic effect of FAK silencing in CL1 compared with CL2 can be explained by the two-fold stronger activity of FAK signaling in CL1 compared with CL2 (Appendix A).

Silencing of Pyk2 did not affect the cell cycle in CL1 and CL2. However, in CL3, siPyk2 reduced the number of cells in G2/M by 9% (*p* < 0.05), while FAK silencing resulted in an accumulation of cells in G1, a reduction in cells in the S phase, and a nonsignificant reduction in cells in G2/M (Figure 6a). This result indicates that in CL3, Pyk2 has a dominant role over FAK in the stimulation of cell division, and this can be related to the strong (three-fold) activation of Pyk2 phosphorylation in CL3 compared with CL1 and CL2, together with low activation of FAK. These results suggest that both Pyk2 and FAK may be involved in modulation of the cell cycle in glioma cells under basal conditions, leading to the activation of proliferation. The extent of this modulation and the prevalence of one signaling pathway over the other is patient dependent.

MCM increased the percentage of cells in the G2/M phase compared with control in all evaluated patients (by between 25 and 52%, *p* < 0.05; Figure 6a). The reduction in cells in the G1 and S phases and the increase in viability in response to MCM treatment (Figure 5) confirms the stimulatory effect of MCM on glioma cell proliferation. This effect was reversed by silencing of FAK in all investigated cell lines and also partially by siPyk2 in CL1 and CL3. These results demonstrate that MCM promotes cell proliferation by stimulating cell cycle progression through both the FAK and Pyk2 signaling pathways.

To identify the microglial-released factors involved in the stimulation of cell proliferation through FAK and Pyk2 signaling, glioma cells were treated with previously selected cytokines/chemokines, including EGF, IL-6, IL-8, SDF-1α, PDGFα, PDGFβ, or MCM in combination with one of their respective blockers (gefitinib, tocilizumab, reparixin, burixafor, inhibitor III, or DMPQ, respectively). As demonstrated in Figure 6b–e, administration of EGF, IL-6, IL-8, and SDF-1α led to an increased number of cells in the G2/M phase compared with control in all evaluated cell lines, which is consistent with the cell viability results. PDGFα increased mitosis in only CL3 (Appendix A), and PDGFβ increased mitosis in CL2 and slightly in CL3 (Figure 6f). These results are consistent with the significant increase in viability observed in response to these factors (Figure 5b,c).

The stimulatory effect of EGF on mitosis (Figure 6b) was reversed by Pyk2 silencing in CL2 (decreased by 16%, *p* < 0.05) and by FAK silencing in CL3 (decreased by 14%, *p* < 0.05). In CL1, FAK knockdown in the presence of EGF resulted in G2/M arrest, similar to what was observed under basal conditions (Figure 6a). Treatment with gefitinib inhibited the stimulatory effect of the G2/M phase by MCM (decreased by 7% in CL1, 17% in CL2, and 9% in CL3, *p* < 0.05 for each), suggesting a significant contribution of the EGF stimulation of G2/M by MCM. Silencing of Pyk2 and FAK did not significantly add to the inhibitory effect of gefitinib in MCM in CL2 and CL3, indicating that in these cell lines, EGF is a key regulator of Pyk2 and FAK signaling, leading to cell cycle modulation. However, in CL1, both Pyk2 and FAK silencing provided an additive effect to gefitinib inhibition, indicating that in this cell line, factors other than EGF are involved in the Pyk2- and FAK-dependent regulation of the cell cycle. Therefore, these results suggest that the proliferative effect of EGF is mediated by both FAK and Pyk2 in all patients, but to different extents.

As depicted in Figure 6c, IL-6 increased the number of cells in the G2/M phase in all evaluated patients (by between 10 and 45%, *p* < 0.05). This effect was inhibited by siPyk2 in all patients and by siFAK in CL2 and CL3, while in CL1, siFAK resulted in G2/M arrest, as shown under basal conditions (Figure 6a). These results suggest that Pyk2 and FAK are part of the proliferative mechanism of IL-6. Treatment with tocilizumab blocked the MCM-dependent stimulation of the G2/M phase in all evaluated patients, indicating that the release of this cytokine by microglia is critical for stimulation of the cell cycle. Under MCM + tocilizumab conditions, siPyk2 induced a greater inhibition of the G2/M phase compared with tocilizumab alone (by 21%, *p* < 0.05) in CL1, indicating that IL-6 is involved in cell cycle regulation through Pyk2-dependent and -independent signaling mechanisms. siFAK in CL1 and CL2, as well as Pyk2 silencing in CL2, did not have additive effects in the presence of tocilizumab, suggesting that, in CL2, the main route for IL6′s regulation of the cell cycle is through Pyk2 and FAK. An interesting result was observed in CL3, in which siFAK and siPyk2 reduced the inhibitory effect of tocilizumab by 21% (*p* < 0.05), probably as a result of over-inhibition. These results suggest that IL6 is an important regulator of glioma cell proliferation, but the extent of involvement of FAK and Pyk2 signaling in the process is patient dependent.

The stimulatory effect of IL-8 was slightly reduced by both Pyk2 and FAK silencing (decreased by 13%, *p* < 0.05) in CL1, while in CL2 and CL3, Pyk2 and FAK silencing had no effect on IL-8-driven mitosis (Figure 6d), suggesting that these kinases have little or no involvement in the mechanisms of the IL8-dependent activation of mitosis. A role for IL-8 in MCM-driven cell proliferation in CL1 was demonstrated by the inhibitory effect of reparaxin in the G2/M phase compared with MCM alone (decreased by 19%, *p* < 0.05). However, under MCM + reparaxin conditions, the silencing of FAK and Pyk2 induced a greater inhibition of the G2/M phase (decreased by 5% and 15%, respectively, *p* < 0.05), suggesting that IL8 is just a minor contributor to the Pyk2/FAK regulation of the cell cycle.

As shown in the Figure 6e, SDF-1α increased the number of cells in G2/M by 33 and 64% in CL1 and CL3, respectively, compared with control (*p* < 0.05, for each), and did not significantly affect CL2. siFAK and siPyk2 reversed the effect of SDF-1α in CL1 and CL3, suggesting a critical role for these kinases in the promotion of cell proliferation by SDF-1α. In the CL1 and CL3 cell lines, burixafor abolished MCM-induced mitosis. siFAK in the presence of MCM + burixafor resulted in a reduction in G2/M by 16% (*p* < 0.05) while increasing the number of cells in the S phase by 33% (*p* < 0.05) in CL1, suggesting that in this cell line, SDF-1α can affect mitosis through FAK-dependent and -independent signaling pathways. However, no significant additive effects were observed with siPyK and siFAK in CL3 or siPyk2 in CL1 compared with MCM + burixafor. These results suggest that in two of three cell lines, SDF-1α serves as a key regulator of mitosis. In addition, Pyk2 and FAK could serve as a main route of SDF-1α-driven proliferation, and other signaling pathways could also be involved.

As shown in Figure 6f, PDFGβ increased the number of cells in the G2/M phase in CL2 and CL3 compared with control (by 75% and 25%, respectively, *p* < 0.05). The effects of PDFGβ appear to be mediated by FAK and Pyk2 in CL2, because siFAK and siPyk2 decreased the number of cells in the G2/M phase compared with PDGFβ alone (decreased by 9% and 17%, respectively, *p* < 0.05). However, in CL3, the PDGFβ-mediated proliferative effect seems to be Pyk2-independent, because siPyk2 did not diminish the effect of PDGFβ. Treatment with DMPQ reduced the number of cells in the G2/M phase by 10% (*p* < 0.05), while arresting the cells in the S phase (increased by 13% and 36%, for CL2 and CL3, respectively, *p* < 0.05). Under conditions of MCM + DMPQ, siFAK did not show an additive inhibitory effect in CL2, suggesting that this is the main route for PDGFβ in the regulation of mitosis, while siPyk2 increased the G2/M phase in CL2, suggesting an over-inhibition effect. In CL3, siFAK in combination with MCM + DMPQ treatment decreased the G2/M phase further than MCM + DMPQ treatment alone (decreased by 10%, *p* < 0.05), suggesting that other pathways may be involved in this process.

PDGFα induced cell proliferation only in CL3, which is consistent with the significant increase in viability in this cell line (Figure 5c). Indeed, the G2/M phase was increased 52% by PDGFα compared with control (*p* < 0.05, Appendix A). This effect appears to be FAK/Pyk2 dependent because the silencing of both kinases decreased the G2/M phase by 14% and 20%, respectively (*p* < 0.05 for both). Inhibitor III, in the presence of MCM, decreased the number of cells in the G2/M phase by 9% and arrested cells in the S phase (38% increase, *p* < 0.05) compared with MCM alone. Under conditions of MCM + inhibitor III, siPyk2 further decreased the number of cells in the G2/M phase (decreased by 9%, *p* < 0.05), suggesting the involvement of signaling pathways other than Pyk2. By contrast, siFAK stimulated mitosis by 9%, (*p* < 0.05) compared with the MCM + inhibitor III group, probably by over-inhibition.

In conclusion, although there is heterogenicity in the cytokine-dependent pathways involved in cell cycle regulation in glioma, our results support a pivotal role of FAK and Pyk2 in this process. In contrast to PDGFα, PDGFβ, and SDF-1α, whose functions appear to be patient-specific, EGF and IL-6 are implicated in Pyk2- and FAK-mediated glioma cell proliferation in all subjects evaluated.

## 4. Discussion

Microglia appear to play an important role in glioma tumor progression by affecting the tissue microenvironment [8,34]. An important aspect of microglial function is the release of cytokines and factors that promote glioma cell proliferation and invasion [8]. Recently, we demonstrated that the activation of Pyk2 by factors released from microglia is critical for glioma tumor progression [10]. Here, we evaluated the role of cytokines and chemokines released by tumor-infiltrating microglia on Pyk2- and FAK-dependent glioma cell dispersal and proliferation.

We identified microglia-derived PDGFβ, EGF, SDF-1α, IL-6, and IL-8 as factors involved in the activation of Pyk2 and FAK signaling in glioma cells. Microglia treated with glioma-conditioned medium (GCM) from primary cell lines increased the expression of genes encoding PDGFβ, SDF-1α, IL-6, IL-8, and EGF (Appendix A). The strong positive correlation between the Pyk2 and FAK protein levels in glioma and mRNA levels of EGF, PDGFβ, SDF-1α, and IL-8 (Figure 1) is supported by the upregulation of phosphorylated Pyk2 and FAK following treatment with cytokines and chemokines (Figure 2). However, some degree of variability between cell lines was observed in the activation of Pyk2 and FAK signaling in response to the identified factors. Indeed, treatments with IL-6 and EGF activated Pyk2 and FAK phosphorylation in all evaluated glioma cell lines (Figure 2), while the effect of PDGFβ, SDF-1α, and IL-8 was patient dependent. However, elimination of the stimulatory effect of MCM on Pyk2 and FAK phosphorylation in response to treatment with cytokine/chemokine receptor inhibitors, together with significant autocrine expression of IL-8, SDF-1α, PDGFα, and PDGFβ by glioma cells (Appendix A), support an essential role for all identified microglial-derived factors (IL-6, EGF, IL-8, PDGFα, and PDGFβ) in the activation of these kinases. Additionally, the evaluated glioma cell lines showed increases in total Pyk2 and FAK protein expression upon treatment with SDF-1α, IL-6, and EGF, indicating regulation of Pyk2 and FAK signaling at the level of protein synthesis, in addition to the regulation of phosphorylation. Of note, significant increases in FAK expression were observed with burixafor, tocilizumab, reparixin, and gefitinib treatment in the presence of MCM compared with MCM alone in CL1, suggesting a compensatory effect in this cell line due to inhibition of the kinase. Similar results for Pyk2 and FAK have been observed in different cell types (hemopoietic cancer cells [35], ovarian cancer cells [36,37], smooth vascular cells [38], and prostate cancer cells [39]) following treatment with SDF-1α, IL-6, EGF, IL-8, and PDGFβ. Our results expand on those of previous studies indicating that the activation of Pyk2 and FAK in gliomas is induced by microglia-derived EGF, PDGFβ, SDF-1α, IL-6, and IL-8.

It was previously reported that Pyk2 and its homolog, FAK, act as critical mediators for activating signaling pathways that regulate cell migration, proliferation, and survival in a number of cell types, including gliomas [40]. Studies by Lipinski et al. [24] demonstrated a differential role of Pyk2 and FAK in terms of glioma migration and proliferation. Specifically, FAK was shown to be involved in glioma cell proliferation [25,26], while the increased activity of Pyk2 correlated with invasion by glioma cells [10,27]. At variance with these studies, our results indicate that both invasion and proliferation are cytokine-dependent, Pyk2- and FAK-mediated processes. Although the reason for this discrepancy is unknown, it could be related to the fact that our study employed primary cell lines from human glioma tumors, while commercial cell lines were used in the earlier studies. It is noteworthy that primary cell cultures have gradually replaced commercial cell lines due to their better representation of in vivo cancer cell behaviors [41,42].

To migrate and invade surrounding tissues, cancer cells, including glioma cells, develop protrusions, known as invadopodia, that degrade the ECM through the release of metalloproteases [24,27]. Our results showed that MCM induces functional invadopodia formation and that this process is Pyk2 and FAK dependent in all evaluated cell lines except for CL3, which depended on Pyk2 only (Figure 3). These results suggest that microglia release factors that activate Pyk2 and FAK and promote glioma cell invasion through matrix degradation. PDGFβ and SDF-1α stimulated the activity of invadopodia in all evaluated cell lines. This effect was abolished by siPyk2 and siFAK, except in CL3, which was Pyk2 dependent only. However, despite the fact that the number of cells with invadopodia was also increased by PDGFβ and SDF-1α in the CL1 and CL3 cell lines, it was unaffected in CL2. CL2 exhibited an increased invadopodia formation at basal levels, explaining the lack of stimulation by these cytokines. Of note, siPyk2 and siFAK in the presence of PDFGβ and SDF-1α abolished the effect of these cytokines on the invadopodia activity in CL1 and CL2, but not in CL3, for which this process seems to be Pyk2 dependent only. Treatment with DMPQ or burixafor in the presence of MCM decreased the invadopodia activity in all evaluated cell lines, confirming that the release of microglial PDGFβ and SDF-1α promotes glioma cell invasion. These results suggest that PDGFβ and SDF-1α are the primary activators of Pyk2 and FAK that induce glioma cell matrix degradation. Our findings are consistent with studies by Chen et al. [43] in which CXCL12 (also known as SDF-1α) induced invadopodia formation and invasion in U87MG cells through the activation of cortactin (a protein associated with ECM degradation) and the regulation of actin polymerization. Furthermore, it was recently shown in breast cancer cells [44] that the cortactin-mediated mechanism is promoted by the activation of Pyk2. Similar to the results of our study, these reports indicated that FAK regulates tumor cell invasion. However, FAK involvement occurs through FAK-regulated focal adhesion-mediated motility, while Pyk2 promoted tumor cell invasion by controlling invadopodium-mediated functions. Similar mechanisms activated by microglia-derived cytokines could be employed in our study, as we observed that Pyk2 knockdown in CL2 and CL3 resulted in matrix tearing and deformation (Figure 3g,m), suggesting different mechanisms of cell invasion by FAK and Pyk2. This result is consistent with studies by Fraley et al. [45] demonstrating that focal adhesion proteins, including FAK, regulate the 3-D migration of cells through the ECM by affecting protrusion activity and matrix deformation. Altogether, these studies demonstrate that Pyk2 and FAK are activated mainly by microglia-derived PDGFβ and SDF-1α and thereby induce glioma functional invadopodia formation in glioma cells.

Cell migration studies support the role of microglia-derived PDGFβ, EGF, and SDF1α in Pyk2- and FAK-dependent glioma dispersal: all evaluated cell lines demonstrated enhanced migration under the influence of MCM, as well as PDGFβ or EGF, and this effect was inhibited by Pyk2 and FAK knockdown. Treatments with DMPQ and gefitinib reversed the MCM-induced migratory action, supporting the role of microglia-derived PDGFβ and EGF. SDF-1α stimulated migration in CL1 and CL3, but not in CL2. However, burixafor reversed the migration-promoting effect of MCM in all investigated cell lines. Taking into account the strong autocrine expression of SDF-1α in CL2 (Appendix A), we can state that SDF-1α in the glioma environment, together with EGF and PDGFβ, are the main regulators of glioma cell dispersal. The additive effects for the inhibition of cell migration were observed in siPyk2 and siFAK knockdown compared with control cells treated with burixafor or gefitinib in the presence of MCM in CL1. This suggests that in CL1 signaling, mechanisms other than Pyk2 and FAK are involved in SDF-1α- and EGF-promoted cell migration. Our results are consistent with previous reports indicating the activation of glioma cell migration by Pyk2-dependent [10,24] or FAK-dependent [46,47] pathways. Therefore, the findings reported in this study demonstrate that PDGFβ, EGF, and SDF-1α are the main effectors of glioma cell migration through Pyk2 and FAK activation.

Studies performed by Lipinski et al. [24] in T98g, SF767, U118, and G112 glioma cell lines demonstrated that cells with high endogenous proliferation rates exhibit low migratory activity and that this effect depends on the status of Pyk2 and FAK activation. Our results confirmed the view that MCM induces both proliferation and migration rates. However, this effect appears to be patient dependent. Specifically, CL3 exhibited a high increase in migration rate (4.3-fold vs. control, Figure 4e) and a small increase in proliferation (64% vs. control, Figure 5c) in response to MCM treatment, while CL1 and CL2 demonstrated significant increases in both migration and proliferation in response to MCM treatment.

In addition, previous studies have shown that microglia and macrophages release cytokines that regulate cell cycle pathways, contributing to glioma progression [48,49]. Consistent with these studies, we demonstrated that MCM induces an intense mitotic stimulation in all evaluated cell lines (a 25–52% increase in the G2/M phase vs. control), indicating that microglial soluble factors induce glioma cell proliferation (Figure 6a). We found that EGF, IL-6, and IL-8 significantly increased the number of cells in the G2/M phase in all investigated cell lines, while SDF-1α affected mitosis in CL1 and CL3, and PDGFβ in CL2 and CL3 (Figure 6b–f). Among the factors identified, EGF, PDGFβ, SDF1α, and IL-6 were found to regulate mitosis, mostly through Pyk2 and FAK, while in the IL-8-dependent regulation of mitosis, mechanisms other then Pyk2 and FAK signaling are involved. Additionally, the role that Pyk2 and FAK play in mitosis varied for each cell line. In CL1, the effects of EGF and IL-6 are mostly Pyk2 dependent. In CL2, EGF mostly affected proliferation through Pyk2, but IL-6 depended on both Pyk2 and FAK mechanisms. In CL3, the effect of EGF was FAK dependent, while the effect of IL-6 was both Pyk2 and FAK dependent. Of note, the additive effects observed in Pyk2- and FAK-silenced cells in the presence of the inhibitors gefitinib, tocilizumab, burixafor, and DMPQ suggest that multiple MCM factors are involved in the Pyk2 and FAK activation that modulates the cell cycle in glioma cells (Figure 6b–f). Based on these results, we suggest that EGF, IL-6, SDF-1α, and PDGFβ together orchestrate Pyk2 and FAK regulation in glioma cells, leading to cell cycle activation. These results are consistent with previous studies with endothelial [50] and human foreskin fibroblast cells [51,52], indicating that cell cycle progression is a Pyk2- and FAK-dependent process. Therefore, these studies point to the central role of Pyk2 and FAK in the microglia-derived EGF and IL-6 induction of glioma cell mitosis and cell division.

The major challenge in treating cancers, including GBM, is heterogeneity from patient to patient and even from cell to cell within the same tumor. For this reason, the identification of pharmacological targets, which aid in overcoming tumor heterogeneity, is significant in cancer research. In this study, we identified key microglia-derived cytokines and chemokines that are involved in supporting tumor progression through Pyk2 and FAK signaling. Considering the individual variability in microglial cytokines’ profiles and the tumor cell response capacity to specific cytokines, Pyk2 and FAK represent a promising target to eliminate the microglial effect on tumor progression.

## 5. Conclusions

Our findings demonstrate that cytokines released by microglia activate Pyk2 and FAK kinases to promote glioma cell proliferation and dispersal. We identified microglia-derived EGF, PDGFβ, SDF-1α, and IL-6 as the primary activators driving Pyk2 and FAK activation in glioma. Specifically, EGF, PDGFβ, and SDF-1α promote glioma cell invasion by inducing the formation of functional invadopodia and stimulating cell migration. EGF and IL-6 together, and to some extent also with SDF-1α and PDGFβ, play a role in enhancing glioma cell mitosis and viability. It is noteworthy that glioma cell invasion and proliferation are Pyk2- and FAK-dependent events, although the relevance of each of these kinases is cell line and cytokine dependent. Our study suggests that the use of Pyk2 and FAK inhibitors may provide a beneficial effect in glioblastoma treatment. Additionally, given the important role of both Pyk2 and FAK in microglia-supported glioma progression and dispersal, this study justifies the use of combined Pyk2/FAK blockers, which may have greater clinical relevance than specific Pyk2 or FAK inhibitors alone.

## Figures and Tables

**Figure 1 cancers-13-06160-f001:**
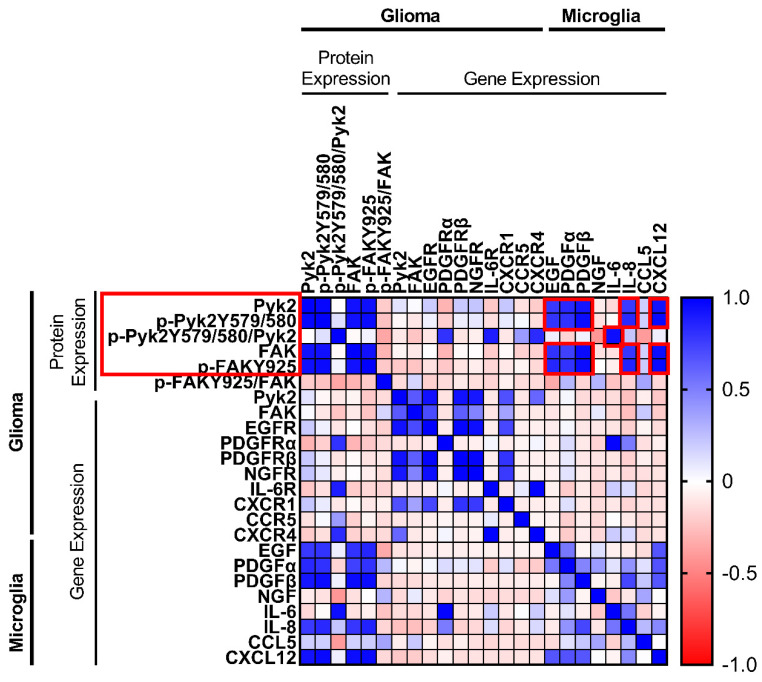
Heat map of Pearson correlation matrix for microglia cytokine and chemokine gene expression and the corresponding cell-surface receptor gene expression, as well as Pyk2 and FAK gene and protein expression in human glioma cells. Cytokines/chemokines with a strong correlation (+0.6–+1.0) selected for further study are highlighted in red (*n* = 20).

**Figure 2 cancers-13-06160-f002:**
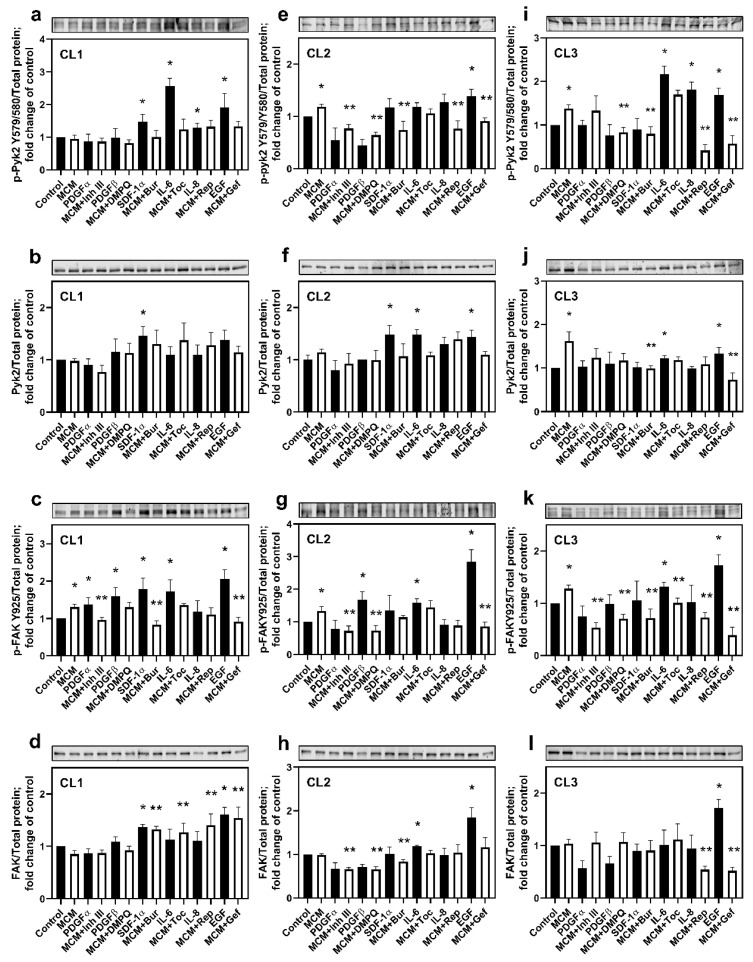
Cytokines and chemokines released by microglia upregulate Pyk2 and FAK protein phosphorylation in human glioma cells. Representative western blots and quantitative results for total and phosphorylated Pyk2 (Y579/580) and FAK (Y925) are presented for the cell lines CL1 (**a**–**d**), CL2 (**e**–**h**), and CL3 (**i**–**l**). The degree of phosphorylation was calculated as the ratio of phosphorylated Pyk2 or FAK to the total loaded protein and normalized to the control for each kinase. Total protein-stained membranes are provided in Appendix A. The values are shown as means ± SD for 3–6 experiments per group. * *p* < 0.05 vs. control; ** *p* < 0.05 vs. MCM. MCM, microglia-conditioned medium; Inh III, inhibitor III; Bur, burixafor; Toc, tocilizumab; Rep, reparixin; and Gef, gefitinib.

**Figure 3 cancers-13-06160-f003:**
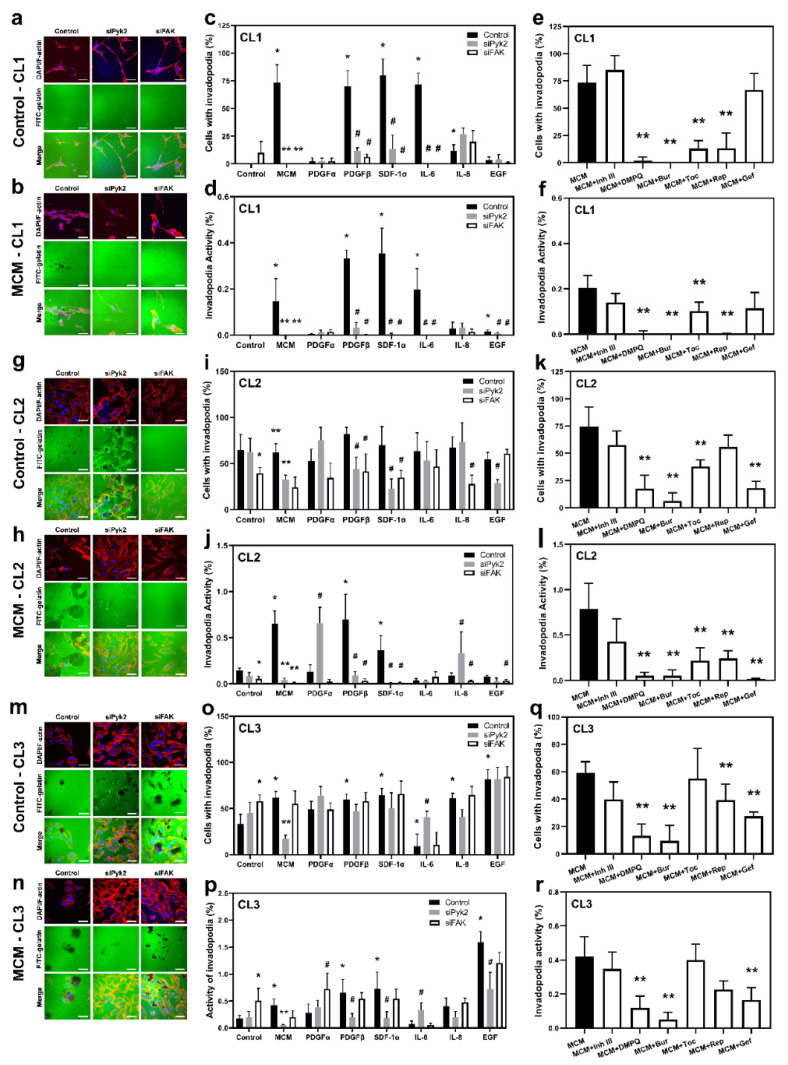
Functional invadopodia formation is enhanced by microglia-derived cytokines and chemokines through Pyk2 and FAK signaling. Fluorescent matrix degradation and invadopodia formation experiments were performed to measure the invasiveness of glioma cells. Representative confocal images of F-actin (rhodamine–phalloidin, red), gelatin (FITC, green), and nuclei (DAPI, blue) under control (**a**,**g**,**m**) and microglia-conditioned medium (MCM; (**b**,**h**,**n**)) conditions are shown for cell lines CL1, CL2, and CL3. Degraded areas of FITC-labeled gelatin are visible as black patches. Oil immersion objectives (40×) were used. Scale bar, 50 µm. The percentage of cells with invadopodia (**c**,**e**,**i**,**k**,**o**,**q**) and invadopodia activity (**d**,**f**,**j**,**l**,**p**,**r**) were calculated for control (black bars), after treatment with cytokines in the presence or absence of siPyk2 (gray bars) and siFAK (white bars) and MCM with or without inhibitors. The values are shown as means ± SD for 3–6 experiments per group. * *p* < 0.05 vs. control, ** *p* < 0.05 vs. MCM, ^#^ *p* < 0.05 vs. the corresponding cytokine. Inh III, inhibitor III; Bur, burixafor; Toc, tocilizumab; Rep, reparixin; and Gef, gefitinib.

**Figure 4 cancers-13-06160-f004:**
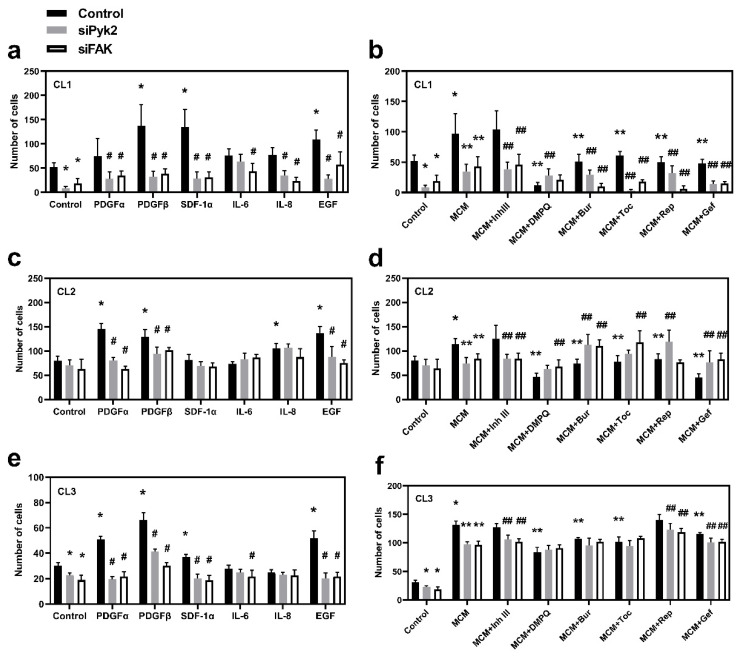
Cytokines released by microglia induce glioma cell migration through Pyk2 and FAK signaling. A Transwell invasion assay was performed to evaluate the migration activity of glioma cells. The number of migrated cells was calculated after treatments with cytokines in mock cells (control) and cells transfected with siRNA against Pyk2 or FAK, as well as cells treated with microglia-conditioned medium (MCM), with or without inhibitors in cell lines CL1 (**a**,**b**), CL2 (**c**,**d**), and CL3 (**e**,**f**). The values are shown as means ± SD for 3–6 experiments per group. * *p* < 0.05 vs. control, ** *p* < 0.05 vs. MCM, ^#^ *p* < 0.05 vs. the corresponding cytokine, ^##^ *p* < 0.05 vs. the corresponding inhibitor. Inh III, inhibitor III; Bur, burixafor; Toc, tocilizumab; Rep, reparixin; and Gef, gefitinib.

**Figure 5 cancers-13-06160-f005:**
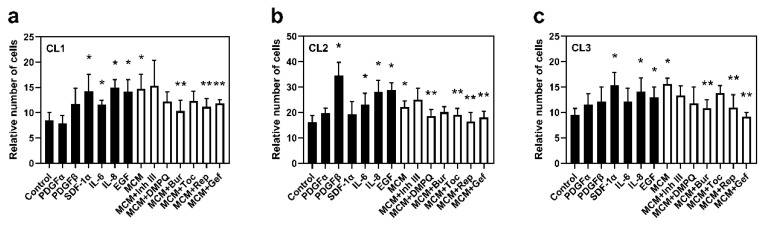
Cytokines released by microglia stimulate glioma cell viability. A trypan blue exclusion test was performed to evaluate glioma cell viability. The number of viable cells was calculated after treatments with cytokines and microglia-condition medium (MCM), with or without the corresponding cytokine inhibitors, in the cell lines (**a**) CL1, (**b**) CL2, and (**c**) CL3. The values are shown as means ± SD of 3–6 experiments per group. * *p* < 0.05 vs. control, ** *p* < 0.05 vs. MCM. Inh III, inhibitor III; Bur, burixafor; Toc, tocilizumab; Rep, reparixin; and Gef, gefitinib.

**Figure 6 cancers-13-06160-f006:**
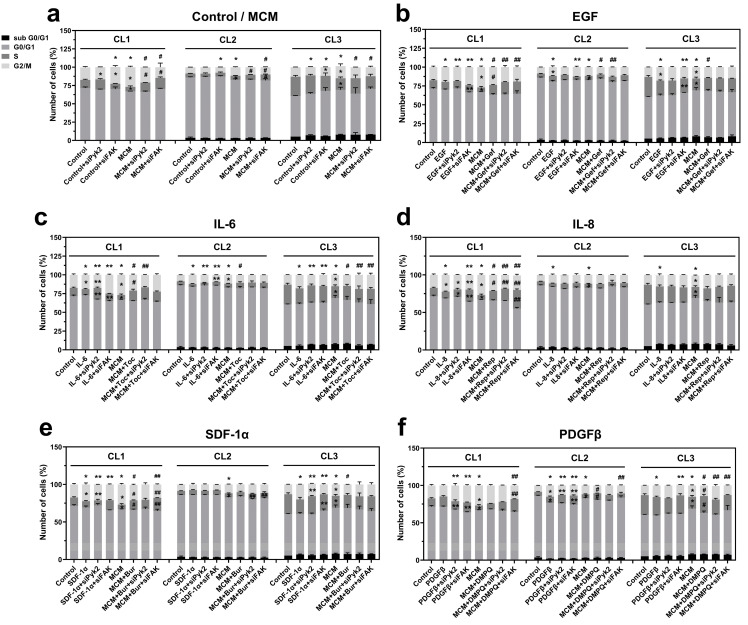
Glioma cell mitosis is stimulated by microglia-derived cytokines through Pyk2 and FAK signaling. Cell cycle analysis was performed to evaluate glioma cell proliferation with flow cytometry. The percentage of cells in the sub-G0/G1, G1, S, and G2/M phases was determined by DNA content in (**a**) control and microglia-conditioned medium (MCM)-, (**b**) EGF and gefitinib-, (**c**) IL-6 and tocilizumab-, (**d**) IL-8 and reparixin-, (**e**) SDF-1α and burixafor-, and (**f**) PDGFβ and DMPQ-treated CL1, CL2, and CL3 cells silenced for Pyk2 and FAK expression. The values are shown as means ± SD for 3–6 experiments per group. * *p* < 0.05 vs. control, ** *p* < 0.05 vs. MCM, ^#^ *p* < 0.05 vs. the corresponding cytokine, ^##^ *p* < 0.05 vs. the corresponding inhibitor. Inh III, inhibitor III; Bur, burixafor; Toc, tocilizumab; Rep, reparixin; and Gef, gefitinib.

## Data Availability

The data generated in this study are available within the article and its Appendix A.

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
