# Peer review of "Microglial Cytokines Induce Invasiveness and Proliferation of Human Glioblastoma through Pyk2 and FAK Activation"

_cancers, 2021, doi:10.3390/cancers13246160_

Round 1

Reviewer 1 Report

Nunez et al aimed to identify the link between tumor-infiltrating microglia and activation of Pyk2- and FAK-dependent glioma cell proliferation and invasiveness that is induced by microglia. The question is of potential interest for the mechanisms by which microglia promote malignancy of glioma cells. However, the results presented in the manuscript fail to support the hypothesis. Below are the major comments:

  1. The heatmap in figure 1 is not appropriately presented and it seems that the genes and their corresponding categories (gene/protein expression and microglia/glioma) are same on both the axes presented. It is not possible to interpret anything from the heatmap.
  2. For all the experiments, authors have presented results for each of the three cell lines (CL1-3) with each cytokine separately, which makes it difficult to understand and interpret them, especially as the results are not consistent between lines. Authors should focus on important results shared by lines because for any results to be significant and generally applicable to glioma, they need to be consistent among different cell lines.  

Reviewer 2 Report

In this paper the authors demonstrated that microglia infiltrate in gliomas promote tumor growth, invasion and treatment resistance. They analised the role of PyK2 and FAK signaling pathways in regulation of migration and proliferation of glioma cells.

The manuscript describes research results obtained by applying several methodologies and it is well organized and conclusions are well supported by the described results. I suggest only few points to revise:

  1. In figure 3 in the confocal image Dapi-F-actin, indicate the magnification, and add a scale bar.
  2. Add a table with the list of the primers used in RT-PCR. Even if the primers are obtained by commercial companies, a table with the amplified gene name, the primer sequences, the size of the amplified DNA segments, and the related references I think should be useful for the readers.
  3. In lines 195 correct the number of cells (5.0x104).
